

# A review of microscopic analysis of blood cells for disease detection with AI perspective

Nilkanth Mukund Deshpande[1,2], Shilpa Gite[3,4] and
Rajanikanth Aluvalu[5]

[1] Department of Electronics and Telecommunication, Symbiosis Institute of Technology,
Symbiosis International (Deemed University), Pune, Maharashtra, India
[2] Electronics & Telecommunication Department, Dr. Vithalrao Vikhe Patil College of Engineering,
Ahmednagar, Ahmednagar, Maharashtra, India
[3] Department of Computer Science, Symbiosis Institute of Technology, Pune Symbiosis
International (Deemed University), Pune, Maharashtra, India
[4] Symbiosis Center for Applied Artificial Intelligence (SCAAI), Symbiosis International (Deemed
University), Pune, Maharashtra, India
[5] Department of CSE, Vardhaman College of Engineering, Hyderabad, Telangana, India

Corresponding author
Shilpa Gite,
shilpa.gite@sitpune.edu.in

## ABSTRACT

**Background:** Any contamination in the human body can prompt changes in blood cell morphology and various parameters of cells. The minuscule images of blood cells are examined for recognizing the contamination inside the body with an expectation of maladies and variations from the norm. Appropriate segmentation of these cells makes the detection of a disease progressively exact and vigorous. Microscopic blood cell analysis is a critical activity in the pathological analysis. It highlights the investigation of appropriate malady after exact location followed by an order of abnormalities, which assumes an essential job in the analysis of various disorders, treatment arranging, and assessment of results of treatment.

**Methodology:** A survey of different areas where microscopic imaging of blood cells is used for disease detection is done in this paper. Research papers from this area are obtained from a popular search engine, Google Scholar. The articles are searched considering the basics of blood such as its composition followed by staining of blood, that is most important and mandatory before microscopic analysis. Different methods for classification, segmentation of blood cells are reviewed. Microscopic analysis using image processing, computer vision and machine learning are the main focus of the analysis and the review here. Methodologies employed by different researchers for blood cells analysis in terms of these mentioned algorithms is the key point of review considered in the study.

**Results:** Different methodologies used for microscopic analysis of blood cells are analyzed and are compared according to different performance measures. From the extensive review the conclusion is made.

**Conclusion:** There are different machine learning and deep learning algorithms employed by researchers for segmentation of blood cell components and disease detection considering microscopic analysis. There is a scope of improvement in terms of different performance evaluation parameters. Different bio-inspired optimization algorithms can be used for improvement. Explainable AI can analyze

the features of AI implemented system and will make the system more trusted and commercially suitable.

## INTRODUCTION

Blood, the most integral part of the body, is constituted of white blood cells (WBC), red blood cells (RBC), platelets, and plasma. This can be further categorized as; cells and platelets are about 45% of human blood, whereas the remaining 55% is filled by plasma (the yellow fluid in the blood). These components and their physical properties like size, shape, color, and count in the whole blood change due to ingress of any foreign object or micro-organism can lead to any sort of infections.

There are different pathological procedures for the detection of diseases. In most cases, microscopic imaging plays a vital role in predicting and detecting abnormalities and occurrences of diseases within the body. Typically, the health of any person is judged by analyzing different features of blood cells and their counts.

### Why the study needed

Previously, manual methods of blood cells analysis were used by pathologists. This might cause error in disease prediction since manual methods are dependent on experience and skills of pathologists. Hence, it is proposed that an automated system of image processing be developed using different algorithms. A simplified, automated and cost effective system is required for detection of diseases. Thus, the above components explained are analyzed for knowing health indication of human being and thereby detecting abnormalities, if any. Although many researchers contributed in the study, there is a need to explore the research in many perspectives.

1. Segmentation of different blood components is still having some shortcomings, such as overlapping cells during the staining.
2. There are different parasitic components in blood cells those need to be identified. So that an existence of a particular malady could be highlighted.
3. There are many challenging diseases like leukemia that have many sub-types depending upon the cell morphology. To detect the exact type of leukemia is still challenging.
4. In medical imaging, the use of artificial intelligence will have a critical issue that it is used as a black box. Hence, it could not be considered full proof and trusted at all times. The technique known as explainable artificial intelligence is the need of study in relation to these analysis concepts.

## Who it is intended for

It is always a critical and crucial job for diagnosing diseases in the medical field Since these decisions are further related to a patient's life. To provide significant contributions in the current diagnostic system is to be intended in many ways. This area is to be studied by a variety of disciplinary and inter-disciplinary researchers. Following are details that will show to whom the study is intended for:

### Academic researchers

Artificial intelligence, machine learning, and deep learning are prime research areas in academics. A disease detection system's performance utilizing microscopic imaging could be improved by applying these algorithms to the current system.

### Doctors

For diagnosis of diseases, doctors rely on the analysis of blood in many cases, in addition to the symptoms of a patient. Blood cell analysis proves to be the most accurate method of diagnosis in the medical field in most disorders. This study provides diagnostic assistance to doctors for further treatment of the patients.

### Pathologists

Blood cells analysis and diagnosis is the leading job of a pathologist. He is responsible for statistical analysis of blood, urine, etc. In some instances, the morphology of blood is important, which is analyzed by microscopic methods. The predictions are dependent upon the experience and skill set of the pathologist in some critical diagnostic conditions. The automated and sophisticated methods of diagnosis via microscopic analysis will prove an assisted diagnostic system for them.

### Commercial use

Pathological analysis is either equipment-based, chemical-based, or morphology-based. Currently, equipment-based systems are costlier, and there is a need to develop a cost-effective automated system for the same. Morphology-based systems can be studied and employed for commercial use that will prove to be cost-effective.

### Bioinformations

A bioinformatician is a person with research in biology, medicine, and health-related studies. These fields are studied with information technology. It involves the collection and interpretation of data covering a range of fields, including genetics or pharmaceutics.

### Haematologics

Hematology is the science that deals with the study of blood and its related disorders. There are some methods related to blood disorders that contribute a fair amount of suggestive diagnosis in hematology.

### Machine learning experts

A machine and deep learning are the branches that prove to be the future of technology in the medical field. There are different methods of machine and deep learning those could

improve the decisions in medical imaging. This article provides the guidelines and challenges towards the utilization of machine learning in macroscopic imaging.

### Technicians in laboratories

The use of AI in laboratories can guide the technician with less experience.

## SURVEY METHODOLOGY

Different approaches are used for searching the articles.

1. Articles are searched through keywords on one of the popular platforms, Google Scholar. The popular keyword considered are white blood cell, red blood cell, disease, machine learning, deep learning, image processing, and explainability.
2. The search is refined by re-arranging the keywords to make the search article specific in-lined with the area. Papers are considered only from the English language.
3. After getting a large number of articles, their abstract is read to finalize them for the reviewing process.
4. Some papers are finalized for the review; those proved to be significantly contributed to the research subject.
5. Cross-references are also searched by scrutinizing some papers having good research contributions.

The following Fig. 1 shows the co-occurrences of different keywords, considering 02 keywords per article as a threshold. The co-occurrence is analyzed by VOSviewer 1.65.

## REVIEW OVERVIEW

This overview is divided into different sections. The first section includes some terminologies related to blood analysis- blood composition and staining process. The second section comprises the microscopic imaging applied to various diseases and their analysis. This section is followed by the generalized methodology for the detection of disease by utilizing image processing and computer vision algorithms. The methods and algorithms adopted by different researchers are discussed in this section. A note on publicly available databases is presented further.

### Some basic terminology related to blood

#### Composition of blood

Figure 2 shows the details of different blood components. Blood is made up of the following elements- erythrocytes, known as red blood cells (RBC), leukocytes, known as white blood cells (WBC), and platelets. These are combinedly present within the plasma membrane. Leukocytes are further classified into two subcategories called granulocytes which consist of neutrophils, eosinophil and basophils, and agranulocytes, consisting of lymphocytes and monocytes. Blood plasma is a mixture of proteins, enzymes, nutrients, wastes, hormones, and gases. Platelets are small fragments of bone marrow cells. The primary function of red blood cells is to carry oxygen from the lungs to different body organs. Carbon dioxide is carried back to the lungs, which will be exhaled afterward. RBC

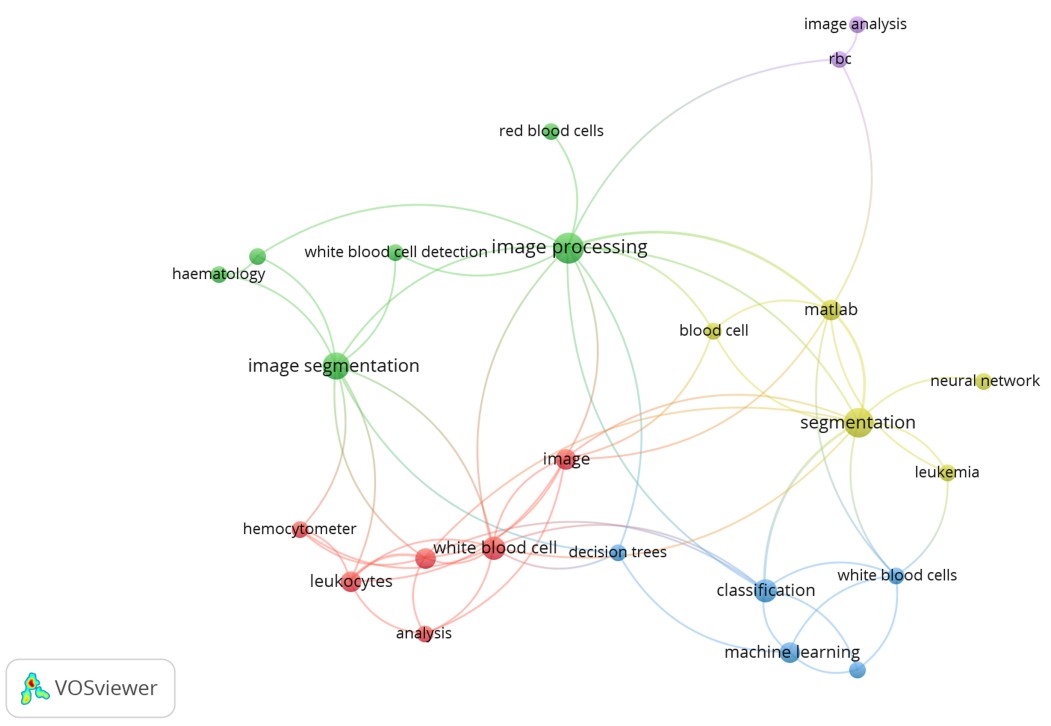

**Figure 1** Co-occurrence of keyword. Source: VOSviewer 1.65.

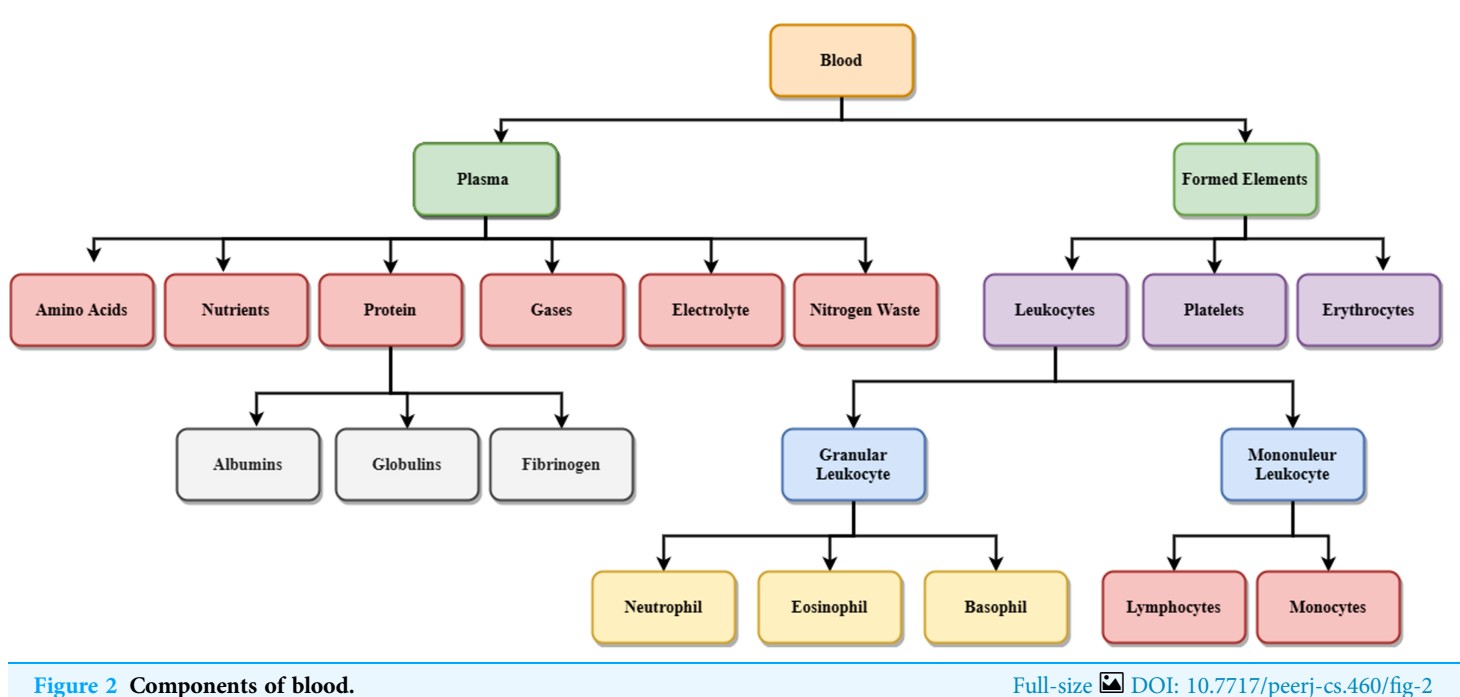

**Figure 2** Components of blood.

count is the measure of different diseases in the human body. A low RBC count means anemia and high means polycythemia. White blood cells protect the body against infection. The different components of blood are identified to know about the health of a human being. Microscopic images of blood smear are analyzed for different disease detection.

### Staining of blood smear

For analysis of microscopic blood images, the blood film needs to be prepared. Glass slide is used for making the blood film. For examination and analysis of this film under the microscope, staining is required. Preparation of blood film requires a slide, a tube, and a blood spreader. Generally, weldge method is used for this purpose. On a base slide, a drop of blood is placed. A spreader slide is moved over this blood drop backward to touch the blood to get the blood spread over the slide uniformly. To get perfection and accuracy in the smear, the spreader slide should be inclined at an angle of 30 degrees to 45 degrees to the blood base slide. The prepared blood smear is dried using an air dryer, and then staining is performed. The dried smear is fixed by absolute methanol or ethyl alcohol. Afterward, it is stained using any of the staining methods—rewmanosky stain, leishmon stain, may-grawald giema or wright-giemsa stain, which differs in the liquid used for staining purpose. These stained slides are then used for analysis under the microscope (*Houwen, 2002*; *Nwogoh & Adewoyin, 2014*; *Vives Corrons et al., 2004*). Generally, laboratories are used to their respective homebrew technics, and therefore, as peripheral blood differential is aphenotypical method, technicians and doctors but also machine learning tools may have problems translating their experience for other laboratories.

## Different areas where microscopic blood analysis is used

Blood smear images are observed under a good quality microscope and are investigated to identify any contamination inside the body (*Poostchi et al., 2018*; *Al-Tahhan et al., 2019*). Several abnormalities could be recognized from these microscopic images. Though there are different laboratory analysis techniques of blood examination available, image processing and computer vision could play a sound and satisfactory role in identifying maladies, preferably it analyzes the morphological abnormality of different components of blood (*Razzak, 2017*; *Loddo, Ruberto & Putzu, 2016*). The following are various areas where image processing and computer vision could be utilized for blood cell analysis (*Othman, Mohammed & Ali, 2017*; *Alom et al., 2018*; *Lavanya & Sushritha, 2017*; *Xia et al., 2019*).

### Blood cell counts

RBC and WBC counts are characteristics of a patient's well-being. It is seen that in a large portion of the cases, the absolute blood cell count is influenced because of an infection within the body. Typically machines are present to count the blood cells and other components in the blood. Nevertheless, when required to get certain particular kinds of observations and abnormalities, there is a need for microscopic analysis. Also, the counting of RBC and WBC is possible by automated computer vision methods considering the cell morphologies. The blood smear is formed after staining that outcomes the film of blood.

These films are observed under the microscope, and the photographs of these images are used for counting. The cells are preprocessed after the microscopic photograph and then segmented to get the required region of interest for counting. Image processing and computer vision methods are utilized for counting purposes. Isolated blood cells are counted via automated algorithms rather than manual, which enhances accuracy (*Abbas, 2015*; *Bhavnani, Jaliya & Joshi, 2016*; *Miao & Xiao, 2018*).

### Detection of viral diseases such as malaria, dengue, chikunguniya, hepatitis

The decrease in RBC and platelets are observed during viral infections. Moreover, the infections' parasites are also identified to realize the viral ailment like intestinal sickness, dengue, chikungunya, or hepatitis. Pathologist distinguishes these contaminations by microscopic blood cell analysis. This process of identification of different parasites is done by automated techniques involving computer vision, image processing, and machine learning algorithms (*Poostchi et al., 2018*).

### Leukemia detection

Harmful reasons, such as leukemia, seriously influence the body's blood-forming tissues and lymphatic framework. In leukemia, the white blood cells created by bone marrow are anomalous. Figure 3 shows the distinction between normal and leukemia cells. It has two significant subtypes, acute leukemia, and chronic leukemia. Leukemia can further be classified into other following types, namely, acute lymphocytic (ALL), acute myelogenous (AML), chronic lymphocytic (CLL), and chronic myelogenous (CML). Recognition of these malignant growth cells is done manually by microscopic image analysis and requires a competent pathologist. An improved automated system of leukemia detection is designed based on image processing and machine learning techniques, which ends up being proficient when contrasted with manual detection (*Negm, Hassan & Kandil, 2018*; *Chin Neoh et al., 2015*; *Singh & Kaur, 2017*; *Shafique et al., 2019*; *Putzu & Ruberto, 2013*; *Jha & Dutta, 2019*; *Moshavash, Danyali & Helfroush, 2018*).

### Anemia and sickle cell detection

Abatement in hemoglobin or absence of solid RBC prompts anemia. It can cause contamination of viral sicknesses and issues identified with relaxing. Anemia detection is mostly done by identifying sickle cells in the blood. These sickle cells have a typical crescent moon shape. These cells are recognized and categorized as ordinary cells and sickle cells via automated algorithms involving computer vision and machine learning (*Bala & Doegar, 2015*; *Elsalamony, 2016*; *Alotaibi, 2016*; *Javidi et al., 2018*).

Figure 4 indicates the different blood cells' components through microscopic examination and Wright stained smear of normal blood. For certain neurological disorders diagnoses such as Alzheimer's and Parkinson's diseases, there are no exact criteria related to clinical ways. For improvements in these kinds of diagnoses emerging metabolomics serves as a powerful technique. This includes a study of novel biomarkers and biochemical pathways (*Zhang, Sun & Wang, 2013*). There is one more challenge to identify proteins from the complex biological network that interact with each other as well as with the cell's environment. Pseudo-Guided Multi-Objective Genetic Algorithm

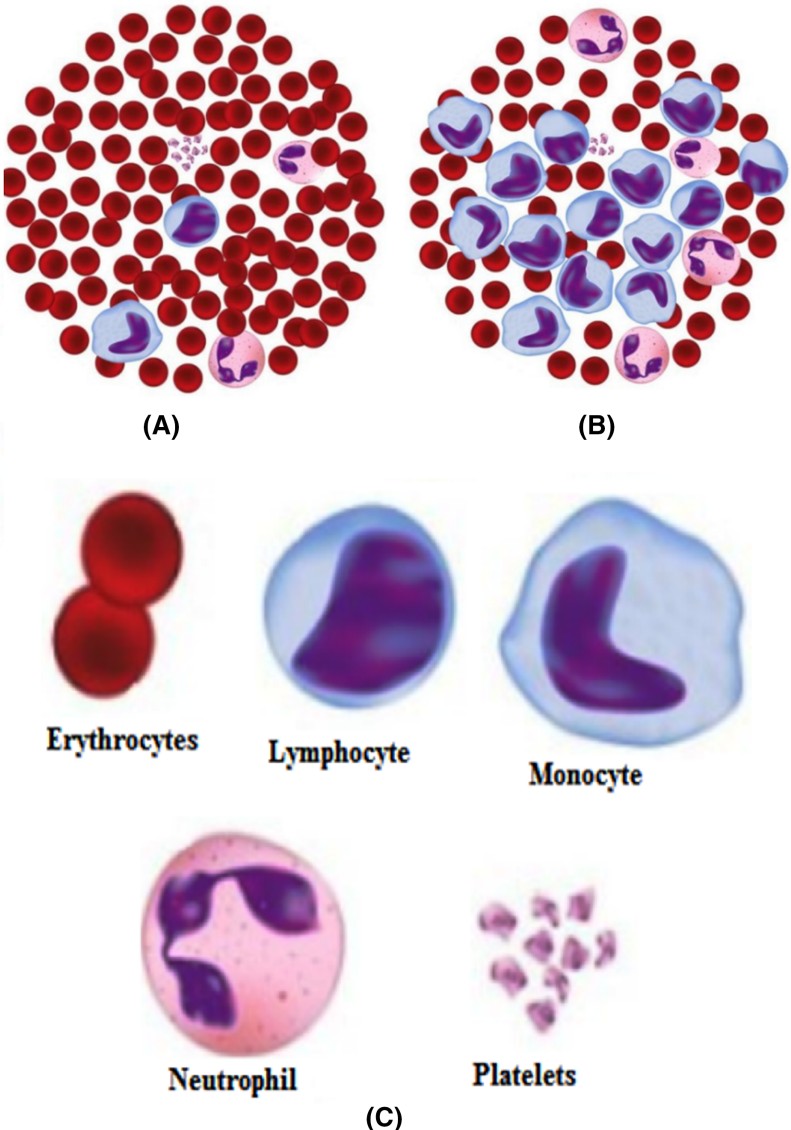

**Figure 3** **(A) Normal cells, (B) Leukemia cells, (C) Different components.**

(PGMOGA) proposed that reconstitutes pathways by assigning orientation weighted network edges (*Iqbal & Halim, 2020*). A gene encoder presented that incorporates two-stage feature selection with an unsupervised type for the classification of cancer samples (*Al-Obeidat et al., 2020*). There is a requirement of finding the correct DNA sequence to get the desired information about the genetic makeup of an organism. A hybrid approach presented utilized Restarting and Recentering Genetic Algorithm (RRGA) with integrated PALS (*Halim, 2020*). For working on the different datasets, there is a need to get a set of visualization metrics. These are used to quantify visualization techniques—an approach of visualization metrics based on effectiveness, expressiveness, readability, and interactivity. Evolutionalry algorithm (EA) is used here as a case study. This methodology can also be utilized further for other visualization techniques (*Halim & Muhammad,*

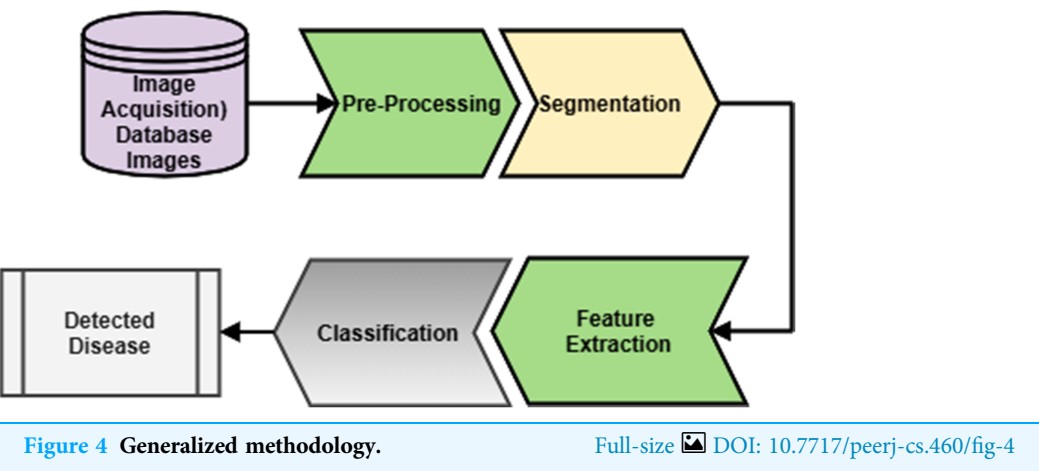

**Figure 4   Generalized methodology.**               

*2017*). There is also a requirement for the extraction of information from larger datasets. A popular Frequent Itemsets (FIs) mining is a task to find itemsets in a transactional database. The graph-based approach is used for the representation of a complete transactional database (*Halim, Ali & Khan, 2019*).

## Paragraph

Out of these different diseases, leukemia is one of the most dangerous in its later stages (*Deshpande, Gite & Aluvalu, 2020*). It develops blasts cells in the blood, which later affect the generation of normal white blood cells. As the number of these blast cells increases, the body gets shortened of healthy cells, leading to frequent infections. Different types of leukemia define the way to treat it. So it is always necessary that the type of leukemia be detected with great accuracy. The morphological differences in the normal and leukemia cells are shown in Fig. 3.

## Generalized methodology

A generalized methodology for microscopic blood cell analysis is shown in Fig. 4. It consists of different stages like image acquisition, image segmentation, feature extraction, and disease detection. The blood sample is taken from the patient by a trained pathologist. After that, a slide is prepared to form a blood smear. The same slide is observed under the excellent quality microscope that will give an image. This image is taken either by camera directly or from an adaptor connected to a microscope. This image is considered for further analysis. Acquired images may have some unwanted regions and overlapping of different components of blood. This image is enhanced by applying a suitable image enhancement technique. So that a good quality image is now available for the analysis. After pre-processing, separation of different blood components is done, including separation of RBC, WBC, plasma, and platelets. By considering the generalized characteristics of blood components, segmentation is done. This process will separate the region of interest for further classification. RBC, WBC and, other components are further classified into their respective sub-classes. This will help specify a particular sub-class image for extracting features for analysis of blood cells, and depending upon the

analysis, detection of disease is done. After the segmentation, different features are extracted by considering different components of blood. Features include size, shape, color, count of different blood components like WBC and RBC counts. Analysis of these features will further detect the disease or count the cells. Depending upon different features extracted, the decision about the disease could be taken. To make the decisions, different classifiers could be designed.

### Image pre-processing

Following are the different methods used for pre-processing. Self dual multi-scale morphological toggle (SMTT) block (*Belekar & Chougule, 2015*), wiener filter (*Patel & Mishra, 2015*), median filtering (*Elsalamony, 2016*; *Bhanushali et al., 2016*; *Mohite Patil & Bhagavan, 2016*; *ANP and Wildlife Service, 1986*; *Thiruvinal & Ram, 2017*), Gaussian filtering (*Yildirim & Çinar, 2019*), gray-scale transformation (*Patel & Mishra, 2015*; *Elsalamony, 2016*; *Bhanushali et al., 2016*; *Mohite Patil & Bhagavan, 2016*; *ANP and Wildlife Service, 1986*; *Bhagavathi & Thomas Niba, 2016*; *Biswas & Ghoshal, 2016*; *Thiruvinal & Ram, 2017*) which has 3 types viz Linear, Logarithmic and Power–law, histogram stretching (*Elsalamony, 2016*; *Bhanushali et al., 2016*; *Mohite Patil & Bhagavan, 2016*; *ANP and Wildlife Service, 1986*), green color component from the RGB image (*Negm, Hassan & Kandil, 2018*), morphological operations (*Elsalamony, 2016*), edge detection (*Biswas & Ghoshal, 2016*).

### Image segmentation

The following are the different segmentation methods employed by the researchers.

Watershed transform (*Belekar & Chougule, 2015*; *Bala & Doegar, 2015*) granulometric analysis and mathematical morphology (MM) operation, fuzzy logic approach (*Bhagavathi & Thomas Niba, 2016*), zack algorithm (*Patel & Mishra, 2015*), K-means clustering (*Negm, Hassan & Kandil, 2018*), marker-controlled watershed segmentation (*Chin Neoh et al., 2015*), stimulating discriminant measures (SDM) based clustering (*Chin Neoh et al., 2015*), Hough transform (*Kaur & Cells, 2015*), iterative thresholding followed by watershed transform (*Biswas & Ghoshal, 2016*), edge thresholding (*ANP and Wildlife Service, 1986*; *Mishra, Majhi & Sa, 2019*) otsu's algorithm (*Thiruvinal & Ram, 2017*; *Poostchi et al., 2018*), a conventional neural network chen prepared Laplacian of Gaussian (LOG) and coupled edge profile active contours(C-EPAC) algorithm (*Poostchi et al., 2018*), triangular thresholding DOST algorithm (*Mishra, Majhi & Sa, 2019*), SMACC, Iterative ISODATA clustering algorithm along with rotation and invariant LBP *Duan et al. (2019)*.

### Feature extraction

There are number of features that could be considered for feature extraction purpose. Some of them are given below

1. **Color Features** Color of the cell can be one of the features which can separate a cell from other types. For example is the color of plasma is very different (yellow) than other blood components. In many cases, the color of the cell talks much about the abnormalities.

2. **Geometric Features** These are the features based on the geometry or shape of the cell. These include following, and like this.

$$Elongation = 1 - \frac{majoraxis}{minoraxis} \tag{1}$$

$$Eccentricity = \frac{\sqrt{major_axis^2 - minor_axis^2}}{minor_axis} \tag{2}$$

$$Rectangularity = \frac{area}{major_axis \times minor_axis} \tag{3}$$

$$Convexity = \frac{Perimeter_convex}{Perimeter} \tag{4}$$

$$Compactness = \frac{4 \times pi \times area}{Perimeter^2} \tag{5}$$

3. **Statistical Features** Statistical moments such as mean and standard deviation gives information about the appearance of distribution. Skewness and kurtosis shape the distribution along with the area and perimeter of the shape. The following are the different statistical features.

$$Mean, \bar{x} = \frac{1}{N}\sum_{i=1}^{N}(x_i) \tag{6}$$

$$Standard\ Deviation, \sigma = \sqrt{\frac{1}{N-1}\sum_{i=1}^{N}(X_i - \bar{x})^2} \tag{7}$$

$$Skewness, SK = \frac{1}{N}\sum_{i=1}^{N}\frac{(x_i - \bar{x})^3}{\sigma^3} \tag{8}$$

$$Kurtosis, K = \frac{1}{N}\sum_{i=1}^{N}\frac{(x_i - \bar{x})^4}{\sigma^4} \tag{9}$$

4. **Texture Features** There are different texture features that are defined such as entropy, correlation, energy, contrast, homogeneity, and so on.
**Entropy** generally defines randomness in the characterization of texture of an image. When co-occurrence elements are same, entropy leads to its maximum value. The equation of entropy as follows.

$$Entropy, Ent = \sum_{i=0}^{N-1}\sum_{j=0}^{N-1}M(i,j)(-\ln(M(i,j))) \tag{10}$$

**Contrast** is the intensity variations in the neighboring pixels in an image.

$$Con = \sum_{i=0}^{N-1}\sum_{j=0}^{N-1}(i-j)^2(M(i,j) \tag{11}$$

**Energy** (E) is the measure of the extent of repetitions of pixel pairs. It gives an uniformity of the image. It gives a larger value for similar pixels.

$$Energy, E = \sqrt{\sum_{i=0}^{N-1} \sum_{j=0}^{N-1} M^2(i,j)} \tag{12}$$

5. **Correlation Features** The repetitive nature of the texture elements position in the image is an important. An auto-correlation function gives the coarseness in an image.

$$Auto - correlation, P(x,y) = \frac{\sum_{u=0}^{N} \sum_{v=0}^{N} I(u,v)I(u+x,v+y)}{\sum_{u=0}^{N} \sum_{v=0}^{N} I^2(u,v)} \tag{13}$$

6. **Inverse Difference Moment or Homogeneity** gauges the local homogeneity of a picture. IDM features acquire the proportions of the closeness of the distribution of the GLCM components to the diagonal of GLCM. IDM has a scope of determining the image and classify it as textured or non-textured.

$$IDM = \sum_{i=0}^{N-1} \sum_{j=0}^{N-1} \frac{1}{1 + (i-j)^2} M(i,j) \tag{14}$$

7. **Directional Moment** In this, the image alignment is considered with respect to the angle.

$$DM = \sum_{i=0}^{N-1} \sum_{j=0}^{N-1} M(i,j)|i=j| \tag{15}$$

### *Classifier for disease detection*

There are different classifiers for the classification of images which are employed for microscopic imaging of blood cells. These include machine learning algorithms as below.

### Decision tree classifier

It falls under the supervised learning type. It is employed for regression as well as classification. It has roots, branches, nodes, and leaves.

Figure 5 shows the different components of the decision tree. The bottom and the upper part are termed roots. A node represents a feature in the dataset. A branch connects two nodes. Different decision tree learning algorithms are there. These include ID3 (Iterative Dicotomizer3), C4.5, C5.0, CART. These algorithms have different characteristics that decide their uses in a particular application.

In the ID3 algorithm, before pruning, trees are extended to maximum size. C4.5 does not require categorical features. It is useful in defining distinct attributes in the case of numerical features. C5.0 has a comparatively smaller rule set and also takes less space than C4.5. The CART classification and regression tree are similar to C4.5; the difference is,

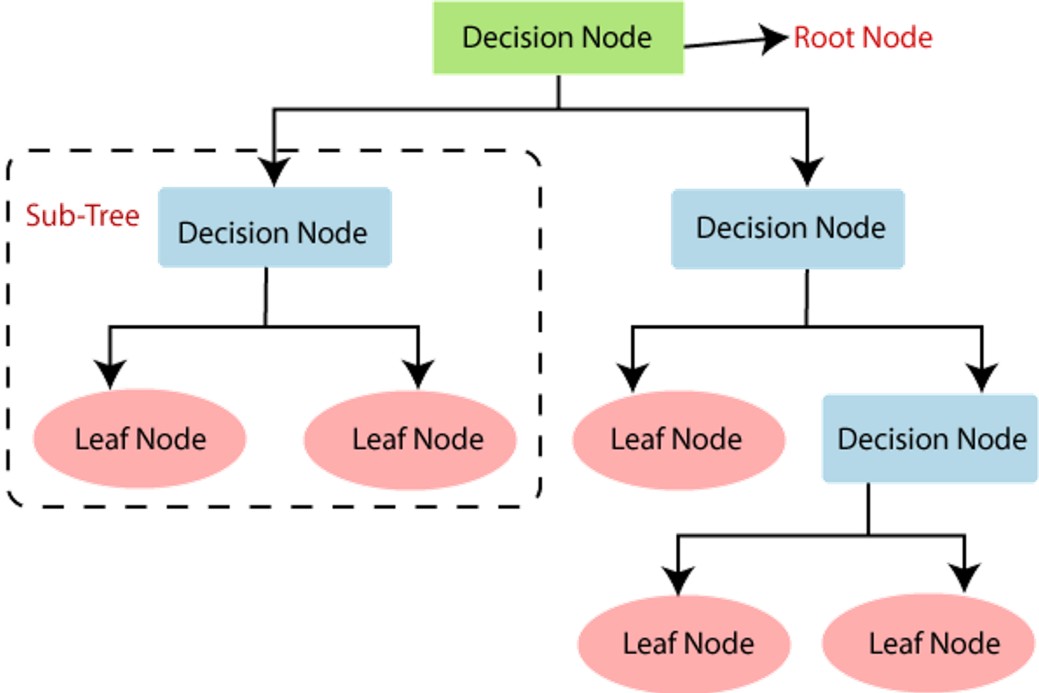

Figure 5 Decision tree. Source: https://pianalytix.com/decision-tree-algorithm/.

it does not calculate the ruleset, and it braces numerical target variables. This type generates a binary tree.

Decision trees may be affected by the problem of overfitting, which could be further analyzed and is taken care of. Regularization is the process to be adapted during the generation of decision trees for compensating for the overfitting issue.

## Random forest

It has many decision trees, with each tree having different training sets. Hence this algorithm is more effective in solving classification problems. An important issue in this algorithm is the selection of a pruning method and branching criteria selection. Popular gain measurement techniques here are the Gain ratio and the Gini index. This algorithm is based on the number of trees to be developed and samples used by the particular node. Figure 6 shows the random forest algorithm.

## K-Nearest Neighbours (KNN)

This classifier employs the nearest instances of training in the attribute space. Here according to the k value of neighbor, the new sample value of the class is decided. For getting the class of the new vector, the closest k samples from training data are selected. There are specific methods for calculating the distances according to the classified samples. These are illustrated as follows.

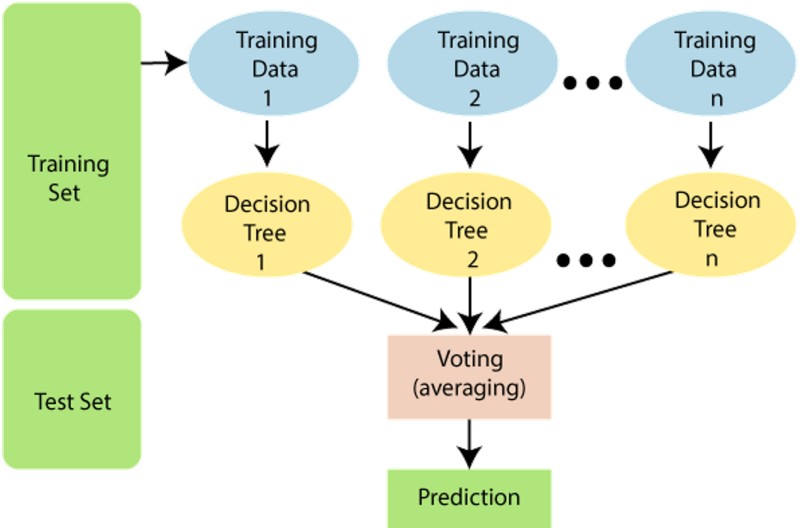

**Figure 6 Random forest algorithm.** Source https://www.javatpoint.com/machine-learning-random-forest-algorithm.

$$Euclidean, \sqrt{\sum_{i=1}^{k} (x_i - y_i)^2} \tag{16}$$

$$Manhattan, \sum_{i=1}^{k} |x_i - y_i| \tag{17}$$

$$Minkowski, \left( \sqrt{\sum_{i=1}^{k} (|x_i - y_i|)^q} \right)^{\frac{1}{q}} \tag{18}$$

## Logistic regression

Regression can be of either linear regression and logistic regressions. Linear regression is a supervised regression algorithm while logistic regression is a supervised classification algorithm. It is categorized into different types. These are binary, multinomial, and the last one is the ordinal. The first type of regression is the,

## Binary logistic regression model

This is the simplest type where the dependent variable can have either 0 or 1 showing only two possible types. So here we have predictor variables multiple but the target variable is binary or binomial.

$$h_\theta(x) = g(\theta^T x), where\ 0 \leq h_\theta \leq 1 \tag{19}$$

g is the logistic or sigmoid function, given as below.

$$g(z) = \frac{1}{1 + e^{-z}}, where\ z = \theta^x \tag{20}$$

A loss function is defined to know the performance of the algorithm using the weights where,

$$h = g(X\theta) \qquad\qquad J(\theta) = \frac{1}{m}.(-y^T \log(h) - (1-y)^T \log(1-h)) \qquad (21)$$

After this, loss function is minimized by decreasing and increasing the weights. The gradient descent tells us how the loss changes with the modification of parameters.

$$\frac{\delta J(\theta)}{\delta \theta_j} = \frac{1}{m} X^T(g(X\theta) - y) \qquad (22)$$

**Multinomial logistic regression** Here 3 or more possible un-ordered types of dependent variables are present with no quantitative significance, such as different types A, B or C. The MLR model is an extension of LR (logistic regression) model. LR Model:

$$\pi(x) = \frac{\alpha + \beta_1 x_1 + \beta_2 x_2 + \cdots + \beta_p x_p}{1 + e^{\alpha + \beta_1 x_1 + \beta_2 x_2 + \cdots + \beta_p x_p}} \qquad (23)$$

*π(x) is an event probability, α represents the dependent variable, β₁, β₂ are independent variables, x₁, x₂... are regression coefficients, p is the number of independent variables and e is the term representing the error.*

From this LR model, MLR can be obtained as an extended version as below. MLR Model:

$$\pi_j(x_i) = \frac{e^{\alpha_i + \beta_1 jx_1 i + \beta_2 jx_2 i + \cdots + \beta_p jx_p}}{1 + \sum_{j=1}^{k-1} e^{\alpha_i + \beta_1 jx_1 i + \beta 2jx2i + \cdots + \beta_p jx_p}} \qquad (24)$$

*here $j_1, j_2..j_k$ are the k categories and n ($i_1, i_2...i_n$) are the possible independent levels.*

## Ordinal regression

It deals with a quantitative significance having 3 or more ordered types for the dependent variable. As an example, variables can be "poor", "good", "very good" or "excellent" categories, can be represented as a score of 0 to 3.

## Naïve Bayes algorithm

It is based on Bayes' theorem and is a probabilistic algorithm. It assumes the occurrence of a feature independent of the other features. Bayes theorem forms its basis,

$$P(Y|X) = \frac{P(X|Y)P(Y)}{P(X)} \qquad (25)$$

It gives the relation between an arbitrary event X with y causing due to some random process. Here P(X) is the input probability, and P(Y) is considered as output probability, while P(Y–X) defines the probability of Y states versus the X input probability.

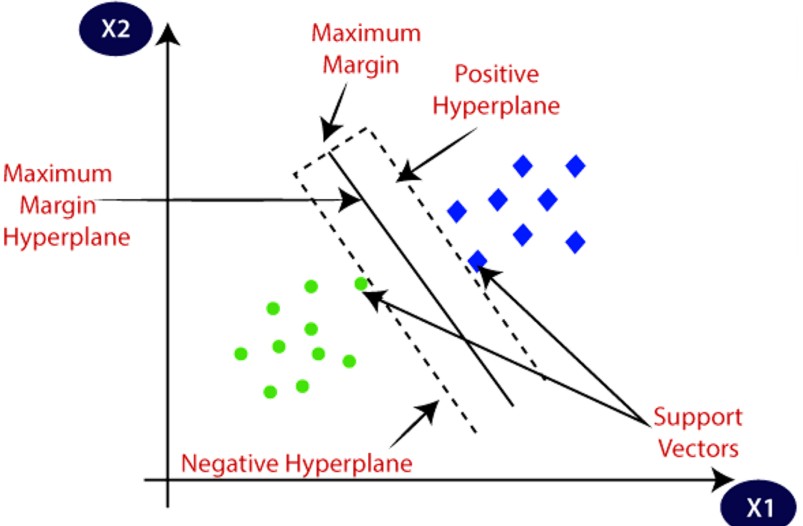

**Figure 7 SVM and its different concepts.** Source: https://www.javatpoint.com/machine-learning-support-vector-machine-algorithm.

The major disadvantage of this algorithm is, it assumes all features as independent features. Hence not possible to get the relationship between the features.

It has three types viz Gaussian, Multinomial, and Bernoulli.

**Gaussian Naive Bayes** In this type, features have values that are continuous in nature and are considered to have Gaussian distribution.

**Multinomial Naive Bayes**: In this case, events are considered to occur in multinoial distribution type. The feature vector is formed by the frequencies of occurrence of some events. Typical use includes the document classification.

**Bernoulli Naive Bayes**: Here, input describes the features with a binary variable independent of each other. In this model, features used are binary term occurrence instead of frequencies.

## Support Vector Machine (SVM)

Support vector machine (SVM) is a machine learning algorithm that has a supervised type. It finds its application in classification as well as regression. It has two types. Linear SVM, used for linearly separable data and non-linear SVM, is used for the data that is non-linearly separable; a non-linear type of SVM is used. Figure 7 explains the SVM and its different concepts. The followings are important terms in SVMs.

Hyperplane is the space or plane for decision. It has a division in a set of different objects having separate classes.

Support Vectors These are the data points which are closest to the hyperplane. These data points define the separating line.

Margin The gap between two lines on Closest data points is considered to define the margin. It is given by the gap in the lines of different classes. The perpendicular distance

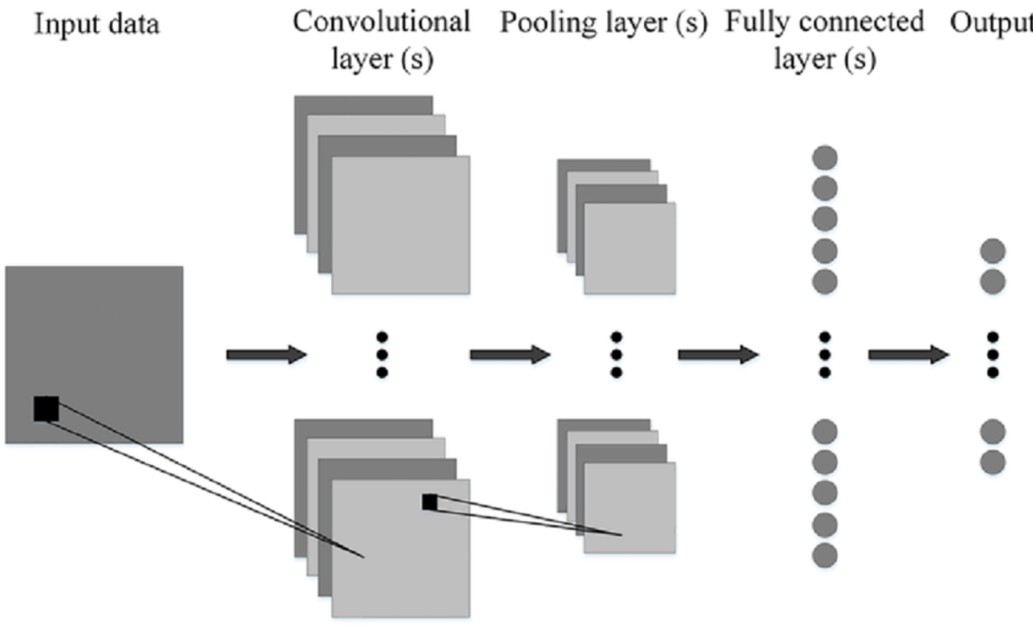

**Figure 8 Generalized CNN architecture (*Phung & Rhee, 2019*).**

between the line and the support vectors is used to find the margin. Larger margins are good, and smaller ones prove to be bad.

## Convolutional neural networks

Figure 8 shows the generalized architecture of convolutional neural network (*Phung & Rhee, 2019*). The CNN has two basic building blocks:

**The Convolution Block**—Consists of two layers, namely, the Convolution and the Pooling Layer. For feature extraction, these layers prove to be the essential components.

**The Fully Connected Block**—has a simple neural network that is fully connected architecture. This layer's function is the classification based on the input coming from the convolutional block.

**Pooling Layer**—It has the process where the extraction of a value is done from a set of values. It generally uses the maximum or the average value. This reduces the output matrix size. There are some popular architectures of CNN given as, Classic network architectures viz LeNet-5, AlexNet, VGG 16, and Modern architectures namely Inception, ResNeXt, ResNet, DenseNet.

### *Publicly available databases*

There are different databases publicly available for the analysis of microscopic blood images. This Dataset contain microscopic images of different parts of blood cells, including white blood cells (WBC) and red blood cells (RBC), along with their sub-classes.

### BCCD Database

Total images in this data are 12,500. Each subtype, Eosinophil, Lymphocyte, Monocyte, and Neutrophil, has approximately 3,000 images. This Dataset also includes additional 410

images of WBC and RBC with JPEG and xml metadata format. This database is under MIT license (*Yildirim & Çinar, 2019*).

**ALL-IDB (Acute Lymphoblastic Leukemia Database)** This database is for the analysis and research on the disease of acute leukemia. This dataset has two subtypes ALL-IDB1 and ALL-IDB2. All images are taken with Canon Powershot G5camera. The microscope has 300 to 500 magnification. Images follow the JPEG format with a color depth of 24-bit. ALL-IDB1 dataset has 109 images with a resolution of 2,592 × 1,944 and a total element of 3,900. It has a total of 510 lymphoblasts. ALL-IDB2 dataset has 260 images with a resolution of 257 × 257 and a total element of 257. It has a total lymphoblast of 130 (*Labati, Piuri & Scotti, 2011*; *Donida Labati, Piuri & Scotti, 2011*).

## Atlas of hematology by Nivaldo Mediros

This database contains different types of microscopic blood cell images. Different image types include maturation sequence, alteration of erythrocytes, anemia images, leukemia images, parasites, fungus images, and some miscellaneous images. The photos of images were taken with the magnification of 200, 400, 360, and 1,000 by the photo-microscopes of Zeiss and Nicon (*Acharya & Kumar, 2019*).

## ASH image bank

This database is from the American Society of Hematology (ASH). It is basically an online image bank of having leukemia cells images. This image library web-based. This offers a wide collection of hematology categories. They provide images with variations in resolutions (*Agaian, Madhukar & Chronopoulos, 2014*).

## Leukocyte Images for Segmentation and Classification (LISC)

This dataset contains images of healthy subjects. The total number of images in the dataset is 400 from 100 different blood slides of healthy eight subjects. The size of the images is 720 × 576. These images are from Hematology-Oncology and BMT Research Center of Imam Khomeini Hospital in Tehran, Iran. A total of 250 images are available for experimentation purposes. Images have different elements of leukocyte like eosinophil, basophil, lymphocyte, monocyte, and neutrophil (*Rezatofighi & Soltanian-Zadeh, 2011*; *Livieris et al., 2018*).

## C-NMC dataset

This database is owned by The Cancer Imaging Archive (TCIA) public access. This database has a total of 118 participants, with a total of 15,135 images. All images are in bit map image (BMP) format. This data is divided into train sets, preliminary test sets, and final test sets with different cancerous and normal images. Table 1 shows the comparison of different publicly available datasets, and Table 2 gives the information of datasets availability along with urls.

Table 3 explores the different methods used by researchers along-with the performance measures, datasets, and a number of images used for disease detection.

**Table 1 Different databases available.**

| Name | | Image formats | Number of images | Color depth | Remark |
|---|---|---|---|---|---|
| BCCD Database (*Yildirim & Çinar, 2019*) | | JPEG, xml, metadata | 12,500 | Not mentioned | Different sub-types of blood cells |
| ALL-IDB (Acute Lymphoblastic Leukemia Database) (*Labati, Piuri & Scotti, 2011*; *Donida Labati, Piuri & Scotti, 2011*) | ALL-IDB-1 | JPEG | 109 (510 lymphoblast) | 24-bit, 2,592 × 1,944 | Cancerous |
| | ALL-IDB-2 | JPEG | 260 (130 lymphoblast) | 24-bit 257 × 257 | Cancerous |
| Atlas of Hematology by Nivaldo Mediros (*Acharya & Kumar, 2019*) | | JPEG | 300 | Not mentioned | Visceral leishmaniasis, cellular simlilarity, morphologic similarities |
| ASH Image Bank (*Agaian, Madhukar & Chronopoulos, 2014*) | | JPEG | 5,084 | Not mentioned | Cancerous and other different types of images |
| Leukocyte Images for Segmentation and Classification (*Rezatofighi & Soltanian-Zadeh, 2011*; *Livieris et al., 2018*) | | (LISC) | 400 (720 × 576) | Not mentioned | Healthy subjects with different sub-types of blood cells |
| C-NMC Dataset | | BMP | 15,135 | Not mentioned | Normal and cancerous images of blood cells |

**Table 2 Different databases available.**

| Name | Remark |
|---|---|
| BCCD Database (*Yildirim & Çinar, 2019*) | https://github.com/Shenggan/BCCD_Dataset, https://www.kaggle.com/paultimothymooney/blood-cells |
| ALL-IDB-1 and 2 (*Labati, Piuri & Scotti, 2011*; *Donida Labati, Piuri & Scotti, 2011*) | https://homes.di.unimi.it/scotti/all/ |
| Atlas of Hematology by Nivaldo Mediros (*Acharya & Kumar, 2019*) | http://www.hematologyatlas.com/morphologicsimilarities/drmedeiros.htm |
| ASH Image Bank (*Agaian, Madhukar & Chronopoulos, 2014*) | https://imagebank.hematology.org/ |
| Leukocyte Images for Segmentation and Classification (LISC) (*Rezatofighi & Soltanian-Zadeh, 2011*; *Livieris et al., 2018*) | http://users.cecs.anu.edu.au/hrezatofighi/Data /Leukocyte%20Data.htm |
| C-NMC Dataset | https://wiki.cancerimagingarchive.net/display/Public/C_NMC_2019+Dataset%3A+ALL+Challenge+dataset+of+ISBI+2019 |

# GAP ANALYSIS

After having reviewed the related literature, the following research gaps are obtained.

1. **Gaps in Segmentation of cells**

   Overlapping cells are not considered at the segmentation stage by many researchers. As many practical cases have the overlapping of cells, during the staining procedure (*Duan et al., 2019*; *Safca et al., 2018*).

2. **Gaps in algorithms and methodology**

   There are different bio-inspired optimization algorithms that are not used by most researchers for detection purposes. The hybrid system of different algorithms of the machine and deep learning also used by limited researchers that may give improved results (*Jha & Dutta, 2019*).

**Table 3 Comparison of different methods for disease detection.**

| Author | Year | Methodology | Performance measure | Database | No. of images |
|---|---|---|---|---|---|
| Patel & Mishra (2015) | 2015 | K-means clustering for detection of WBC. Histogram and Zack algorithm for grouping WBCs, SVM for classification | Efficiency: 93.57 -% | ALL-IDB | 7 |
| Chin Neoh et al. (2015) | 2015 | Multilayer perceptron, Support Vector Machine (SVM) and Dempster Shafer | Accuracy: Dempster-Shafer method: 96.72% SVM model: 96.67% | ALL-IDB2 | 180 |
| Negm, Hassan & Kandil (2018) | 2018 | Panel selection for segmentation, K-means clustering for features extraction, and image refinement. Classification by morphological features of leukemia cells detection | Accuracy: 99.517% , Sensitivity: 99.348%, Specificity: 99.529% | Private datasets | 757 |
| Shafique et al. (2019) | 2019 | Histogram Equalization, Zack Algorithm, Watershed Segmentation, Support Vector Machine (SVM) classification | Accuracy: 93.70% Sensitivity: 92% Specificity: 91% | ALL-IDB | 108 |
| Abbasi et al. (2019) | 2019 | K-means and watershed algorithm, SVM, PCA | Accuracy, specificity, sensitivity, FNR, precision all are above 97% | private | Not mentioned |
| Mishra, Majhi & Sa (2019) | 2019 | Triangle thresholding, discrete orthogonal S-Stransform (DOST), adaboost algorithm with random forest (ADBRF) classifier | Accuracy: 99.66% | ALL-IDB1 | 108 |
| Bhavnani, Jaliya & Joshi (2016) | 2019 | MI based model, local directional pattern (LDP) chronological sine cosine algorithm (SCA) | Accuracy: 98.7%, TPR:987%, TNR:98% | AA-IDB2 | Not mentioned |
| Abbasi et al. (2019) | 2019 | K-means and watershed algorithm, SVM, PCA | Accuracy, specificity, sensitivity, FNR, precision all are above 97% | Private | Not mentioned |
| Sukhia et al. (2019) | 2019 | Expectation maximization algorithm, PCA, sparse representation | Accuracy, Specificity, Sensitivity all more than 92% | ALL-IDB2 | 260 |
| Ahmed et al. (2019) | 2019 | CNN | Accuracy: 88% leukemia cells and 81% for subtypes classification | ALL-IDB, ASH Image Bank | Not mentioned |
| Matek et al. (2019) | 2019 | ResNeXt CNN | Accuracy, Sensitivity and precision above 90% | Private | 18,365 |
| Sahlol, Kollmannsberger & Ewees (2020) | 2020 | VGGNet, statistically enhanced Salp Swarm Algorithm (SESSA) | Accuracy: 96% dataset 1 and 87.9% for dataset 2 | ALL-IDB, C-NMC | Not mentioned |

3. **Gaps in system performance**

   Performance measurement parameters in most of the cases are limited to accuracy only (Patel & Mishra, 2015).

4. **Gaps in detected diseases**

   Many diseases are not assured to be detected and diagnosed with their sub-types (Othman, Mohammed & Ali, 2017) This includes leukemia that has many sub-types. Its sub-types such as L1, L2, L3 are not considered in the case of acute lymphoblastic leukemia in most of the cases. In the case of acute myelogenous leukemia, its different subtypes T1 to T7, are not clearly detected in most of the cases (Miao & Xiao, 2018; Shafique et al., 2019). There is less investigation in terms of stages of

diseases that gives a threshold for determining the severity of the disease for anemia, thalassemia (*Bhanushali et al., 2016*).

5. **Databases**

   Accuracy of different stages of blood cell analysis is tested on a limited database.

6. **Gaps in critical decisions**

   In medical imaging, artificial intelligence and deep learning are employed by most researchers. It is less acceptable in the case of critical decisions because the implementations of these algorithms have a black box. Hence its features can not be analyzed, and there is always a problem of getting the right detection by wrong reasons.

## Discussion of explainable AI for critical analysis

In medical imaging, artificial intelligence is mostly used for detection and diagnosis (*Ahuja, 2019*; *Tizhoosh & Pantanowitz, 2018*). It is generally not been preferred as the final decision about the diagnosis. The main reason for this is, it works as a black box with only input and outputs are known. There might be the occurrence of the right decisions related to the diagnosis, but it might be due to wrong reasons. The algorithms such as decision trees elaborate their implementation to a good extent. However, it limits the parameters such as accuracy of diagnosis, although advanced AI algorithms and deep learning algorithms assure good accuracy in terms of diagnosis but unable to explain the insides of implementation-black box (*Lundervold & Lundervold, 2019*). Hence, explainable AI came into the picture to justify the trust of diagnosis in medical imaging (*Singh, Sengupta & Lakshminarayanan, 2020*). This will analyze the black box's features and characteristics in the implementation of AI or deep learning algorithms. In order to make the system to be used with more trust for commercial purposes for the general public, the explainability will prove to be the most suitable and appropriate.

## CONCLUSION

Blood cell analysis assumes a crucial job in location and expectation of various issues and maladies identified with the person. There are distinctive neurotic strategies for the equivalent by and by, which ends up being exorbitant and furthermore requires a long understanding of the location. Image processing and computer vision strategies are produced for the investigation of blood cells and the discovery of maladies. The microscopic blood cell analysis framework has various stages of being specific, pre-processing, segmentation, feature extraction, classifier, and illness identification. Pre-processing comprises improving the gained picture quality and commotion expulsion. This incorporates gray-scale conversion, thresholding, filtering, histogram stretching, morphological operations. Pre-processed image is portioned to get the locale of interest for further processing. Here WBC and RBC, and platelets are isolated. Distinctive computer vision techniques utilized for segmentation are edge detection, watershed transformation, mathematical morphology, zack algorithm, k-means clustering, SDM, HSV thresholding, otsu's algorithm. There are overlapping cells at the time of staining of blood smear. Expulsion of these overlapping cells at the time of segmentation is a difficult

undertaking. Hough transform evacuates certain overlapping; however, it makes the framework slower. Segmented images are classified by algorithms like SVM, ANN classifier, ELM classifier, circular hough transform. There are various databases accessible for experimentation and investigation of microscopic blood cells, such as BCCD (Kaggle) Database, ALL-IDB1, ALL-IDB2, Atlas of Hematology by Nivaldo Meridos, Leukocyte pictures for division and characterization (LISC), Ash image bank, and C-NMC dataset. There are different application territories where microscopic blood cell examination assumes a crucial job. RBC, WBC count, blood group identification, leukemia detection, sickle cells detection, the partition of various WBC sub-classes, malaria parasite detection, could be performed utilizing complex image processing and computer vision techniques. There is always a need to get proper and trusted diagnoses in the medical science field. Machine learning and computer vision can prove the system of suggestive diagnosis to achieve better accuracy than existing approaches. In a major case, when the morphology is concerned with microscopic imaging, pathologists' experience and skillset always matter. There is a need to have software frameworks utilizing machine learning and artificial intelligence to conquer this problem. Also, there is a problem of explainability of AI algorithms towards the particular diagnostic decisions. For this to take care, AI explainable frameworks could be utilized in the future. For the treatment of different diseases, AI can also put its role in a very trusted manner.

## ACTIVE RESEARCH QUESTIONS AND DIRECTIONS TO FUTURE RESEARCH

After an extensive literature survey, the following research questions were raised, motivating further research in this field.

- RQ.1 How to increase the performance of different stages of microscopic analysis for disease detection?
- RQ.2 How to increase the accuracy of detection of critical diseases such as leukemia and its subtypes by using the morphological diagnosis?
- RQ.3 What is the societal impact of analyzing critical diseases at their early stages and defining the stage of diseases?
- RQ.4 Why is there a need to apply hybridized algorithms for the classification of microscopic cells for disease detection?
- RQ5 What are the different performance measures and validation techniques for analyzing the designed framework's performance?
- RQ6 Which validation techniques are to be employed for system validation of AI implemented systems?
- RQ7 How could AI be trusted and used commercially for medical diagnosis by analyzing the features of the black box in AI implementation?

 Following are the future perspectives of the work to improve the contributions in this area.

A powerful division of white and red cells in minuscule blood smear pictures to meet better precision could be actualized. To conquer overlapping cells issue at the hour of division will likewise end up being a significant extension. Viable feature extraction by utilizing distinctive image transforms will likewise demonstrate to a significant degree. Different optimization algorithms could be utilized efficiently for the classification of blood cells. Different deep learning algorithms may demonstrate productivity and might give high accuracy to various phases of examining blood cells. The designed algorithms must be tasted with various publicly accessible databases for precision. The precision of the calculation should be similar enough with all the databases. Another parameter like vigor can be presented for this reason. The relative accuracy of various databases can be determined. To gauge the exhibition of a framework with various measures such as true positive, true negative, faults, sensitivity, specificity, precision, FI score, J-score in addition with accuracy. The contribution is still needed for various ailments location, such as diabetes, viral diseases such as chikungunya and dengue, anemia diseases such as pancytopenia, thalassemia, and leukemia.

### Funding
The authors received no funding for this work.

### Competing Interests
Rajanikanth Aluvalu is an Academic Editor for PeerJ.

### Author Contributions
- Nilkanth Mukund Deshpande conceived and designed the experiments, performed the experiments, performed the computation work, prepared figures and/or tables, and approved the final draft.
- Shilpa Gite conceived and designed the experiments, authored or reviewed drafts of the paper, and approved the final draft.
- Rajanikanth Aluvalu analyzed the data, prepared figures and/or tables, helped to find out methodology, and approved the final draft.

### Data Availability
This is a literature review; there are no raw data or code.

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
