# Peer review of "A review of microscopic analysis of blood cells for disease detection with AI perspective"

_PeerJ Computer Science, doi:10.7717/peerj-cs.460_

## Round 0.1 · original submission · Major Revisions

Manuscript is well written and sound. However, it needs a major revision as suggested by the reviewers.

Reviewer 1 ·

Basic reporting

The paper presents a needed survey on a trending topic in the domain of Bioinformatics.

Experimental design

The article is well-designed, However, a few shortcomings are mentioned to the authors.

Validity of the findings

Its a survey paper, a few new references are mentioned to be added.

Additional comments

The paper presents a needed survey on a trending topic. Overall, the paper is fine, however, following issues needs to be addressed.
1. I could see a few linguistic issues here and there. The authors are advised to have an independent round of proofreading
2. Availability of datasets is always an issue. The authors must add a section listing the available public datasets for the problem at hand with their URLs.
3. Fig. 4 is too generic plus it gets blurred with zooming. Kindly revise.
4. The authors have missed a few recent and important referenced. Pls. consider adding following
4a. Recent advances in metabolomics in neurological disease, and future perspectives
4b. Orienting conflicted graph edges using genetic algorithms to discover pathways in protein-protein interaction networks
4c. Gene encoder: a feature selection technique through unsupervised deep learning-based clustering for large gene expression data
4d. Optimizing the DNA fragment assembly using metaheuristic-based overlap layout consensus approach
4e. Quantifying and optimizing visualization: An evolutionary computing-based approach
4f. On the efficient representation of datasets as graphs to mine maximal frequent itemsets
4g. Coronavirus disease pandemic (COVID-19): challenges and a global perspective

Reviewer 2 ·

Basic reporting

see below

Experimental design

see below,

Validity of the findings

see below

Additional comments

Deshpande focus on microscopic analysis of blood cells for disease detection using AI. This is an interesting and important paper, several aspect need clarification:
1. In figure two, please do not use the word agranulocytes, this is not a good way to discriminate white blood cells, please follow the WHO classification, book published 2017.
2. For staining such as Romanowsky stain: please also prefer already in this paragraph that laboratories are used to their respective homebrew technics and therefor, as peripheral blood differential is aphenotypical method, technicians and doctors but also machine learning tools may have problems to translate their experience for other laboratories.
3. The paragraph starting with line 138 is not clear, what the authors want to say here? Where do they get this information from? Please also mention that normally machines are doing cell counting of peripheral blood and any unknown result by flagging will start the process of preparing peripheral blood smears. Please also refer to the aspect that (line 146) that there may be several technics to take the pictures not only "segmentation.
4. Figure 3 is only an idea what is going on, how ever one should really try to paint the lymphocytes and the monocytes a little bit more according to reality, as showing here they look much too similar.
5. The sentence line 160 is quite funny, please delete the sentence "it is found to be…" as AML is already killing patients several days after diagnoses, the authors see problems in future areas of interest to devide white and red blood cells on smears, for sure this is a problem, however the reviewer does not see the problem to devide those, but to approach those in different ways with respect to machine learning.
6. It would be also helpful to understand the expected audience and readers for this paper: bioinformations, haematologics, machine learning experts, technicians in laboratories with less experience or personal …, please speculate for the future perspectives in "conclusions" about the unmet medical needs and what machine learning can contribute for better diagnostics and better treatment and also to guide algorithms that follow the investigation of peripheral blood in the next 5 to 10 years.

---

## Round 0.2 · accepted · Accept

According to the reviewer, article is fine.

Reviewer 1 ·

Basic reporting

Fine

Experimental design

Fine

Validity of the findings

Fine

Additional comments

The authors have addressed my concerns.